# Faba Bean Processing: Thermal and Non-Thermal Processing on Chemical, Antinutritional Factors, and Pharmacological Properties

**DOI:** 10.3390/molecules28145431

**Published:** 2023-07-15

**Authors:** Abraham Badjona, Robert Bradshaw, Caroline Millman, Martin Howarth, Bipro Dubey

**Affiliations:** 1National Centre of Excellence for Food Engineering, Sheffield Hallam University, Sheffield S1 1WB, UK; c.e.millman@shu.ac.uk (C.M.); m.howarth@shu.ac.uk (M.H.); b.dubey@shu.ac.uk (B.D.); 2Bimolecular Research Centre, Sheffield Hallam University, Sheffield S1 1WB, UK; r.bradshaw@shu.ac.uk

**Keywords:** faba bean seed, vicine, biological activity, bioactive compounds, polyphenols, antinutritional factors

## Abstract

The food industry, academia, food technologists, and consumers have become more interested in using faba bean seeds in the formulation of new products because of their nutritional content, accessibility, low costs, environmental advantages, and beneficial impacts on health. In this review, a systematic and up-to-date report on faba bean seeds’ antinutrients and bioactive and processing techniques is comprehensively presented. The chemical composition, including the oil composition and carbohydrate constituents, is discussed. Factors influencing the reduction of antinutrients and improvement of bioactive compounds, including processing techniques, are discussed. Thermal treatments (cooking, autoclaving, extrusion, microwaving, high-pressure processing, irradiation) and non-thermal treatments (soaking, germination, extraction, fermentation, and enzymatic treatment) are identified as methods to reduce the levels of antinutrients in faba bean seeds. Appropriate processing methods can reduce the antinutritional factors and enrich the bioactive components, which is useful for the seeds’ efficient utilization in developing functional foods. As a result, this evaluation focuses on the technologies that are employed to reduce the amounts of toxins in faba bean seeds. Additionally, a comparison of these methods is performed in terms of their advantages, disadvantages, viability, pharmacological activity, and potential for improvement using emerging technologies. Future research is expected in this area to fill the knowledge gap in exploiting the nutritional and health benefits of faba bean seeds and increase the utilization of faba bean seeds for different applications.

## 1. Introduction

Faba bean (*Vicia faba* L.) is a cool seasonal crop that belongs to the Fabaceae family, also known as horse or field bean, and it is commonly cultivated as a food and feed for animal usage [1]. Australia, China, Egypt, Ethiopia, Germany, Spain, and the United Kingdom are among the most significant agricultural producers of faba bean (FAOSTAT, 2019). Approximately 43,000 faba bean accessions are held in the GenBank database globally [2]. Various varieties of faba bean seeds, with distinct color ranges and sizes, have been reported in the literature (Figure 1) [3]. The increasing market demand for plant-based ingredients as alternatives to animal-based materials has led to growing attention over the past few decades [4]. Hence, several investigations are being conducted on the sensory attributes of plant-based ingredients such as pea and faba bean [5,6]. In addition, the current disparity between the global food supply and the demand for meat-based products is becoming an important concern due to the recent coronavirus disease outbreak (COVID-19), which affected the food supply chain [7]. Concerns regarding environmental sustainability and the protection of animals, as well as health-related benefits, have also increased the recent interest in plant-based ingredients [8,9]. Whole faba bean contains 20–35% protein, 1–2% fat, 55–65% carbohydrate, 10–15% fiber, and vitamins and minerals such as iron, zinc, calcium, potassium, and magnesium [10,11,12]. The presence of phytochemicals in faba bean ingredients has been suggested to provide numerous health benefits, such as antioxidant properties and the inhibition of enzymatic activity during carbohydrate digestion [13,14]. The oil extracted from faba bean is rich in polyunsaturated fatty acids, mostly linoleic and oleic acids [15].

Owing to its nutritive value and techno-functional properties, faba bean is gaining widespread attention in scientific research (Figure 2) and application in a variety of food products, such as beverages, sausages, and meat analogues [16,17], as an alternative to traditional ingredients such as casein, whey, and wheat protein. Faba bean seeds contain flavonoids, phenolic compounds, and other bioactive compounds that are used to confer several physiological benefits beyond nutrition, such as neuroprotective, anticancer, antioxidant, hypocholesterolemic, and antimicrobial effects [18,19].

The presence of several antinutritional factors in fresh FBS has hindered its utilization in food. The most limiting antinutrients in faba bean are the pyrimidine glucosides vicine and convicine [20]. Vicine and convicine, when consumed, may result in hemolytic anemia in persons with a deficiency in glucose-6-phosphate dehydrogenase [21]. Phytic acid (inositol dihydrogen phosphate (IP6) or inositol polyphosphate) is a chelating agent that binds divalent cations such as iron and interferes with protein absorption through complex formation [22]. Raffinose-series oligosaccharides have also been implicated in digestive discomfort, such as gas production and flatulence, due to the anaerobic fermentation of these carbohydrates. Different faba bean cultivars have significantly different levels of antinutritional substances (tannins, phytic acid, trypsin inhibitors, and saponins) [10]. This limitation can easily be addressed by different processing treatments; thus, faba bean can provide a significant contribution to the human diet. Several processing techniques have been used by scholars to investigate methods to reduce these antinutrients [23,24,25,26,27,28,29]. Due to the huge potential for the incorporation of FBS ingredients into numerous food products, the literature on the impact of processing to reduce these toxins and antinutrients and increase their utilization need to be reviewed.

This review provides a comprehensive summary of the chemical composition, antinutritional factors (vicine, convicine, phytic acid, tannins), and bioactive components (flavonoids, peptides) present in FBS. Further attention is given to the potential of faba bean seeds as bioactive constituents and in peptide preparation due to their therapeutic effects. Attention is also drawn to the impacts of processing techniques to reduce antinutrients and allergic proteins. This review provides researchers with an up-to-date, current understanding of the extraction of bioactive compounds from faba bean and effective means of reducing the toxicity of faba bean antinutrients to tolerable levels.

**Figure 1 molecules-28-05431-f001:**
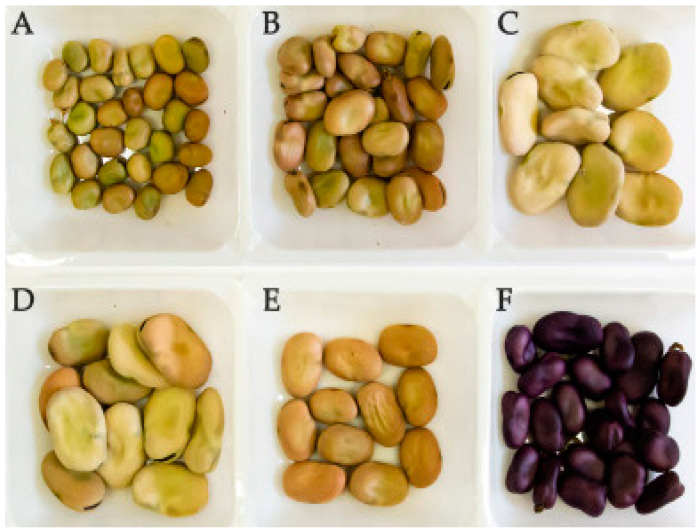
Faba bean seeds with different seed color and size. (**A**) small-seeds; (**B**) medium-seeds; (**C**) large-seeds; (**D**) green seeds; (**E**) brown seeds; (**F**) purple seeds. Reprinted with permission from seed colors and sizes [30].

**Figure 2 molecules-28-05431-f002:**
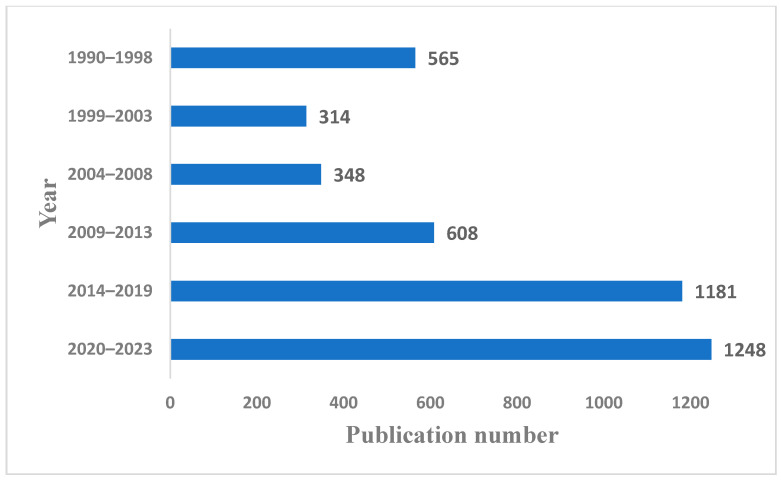
Number of publications reported in the Scopus system containing the term “faba bean” in the title, abstract, and keywords within the years 1990 to 2023 (as of 20 June 2023).

## 2. Faba Bean Seeds’ Constituents and Properties

### 2.1. Proteins

The protein content of faba bean seeds varies from 26.2 to 32.8% [31]. A similar report by [11] showed that whole faba bean had protein content of 31.2 g/100 g, while the dehulled form had protein content of 35.5 g/100 g, higher than that recorded for lupin, rapeseed, quinoa, and buckwheat. The protein content of faba bean flour was observed to be 29.99 g/100 g, higher than that of other pulses such as green pea, yellow pea, and wheat flour, representing 32% of the total energy source, compared to 14.5% for wheat flour [32]. The protein content was found to range between 25 and 27%, with two outlier varieties showing protein content of approximately 32% for 40 varieties of faba bean [11]. The nutrient compositions of faba-bean-derived ingredients such as dehulled seeds, concentrates, and isolates differ in terms of protein, fat, mineral, and carbohydrate content, as shown in Table 1. High protein content with a low carbohydrate ratio is usually observed in isolates and concentrates, compared to whole and hulled seeds. The protein content of isolates was also reported to be 90% [33] higher compared to the corresponding flour and concentrates. Protein isolation from faba bean presents an enormous opportunity as a means to produce ingredients that are virtually free from vicine and convicine [34].

Foods classified as good protein sources are those with 20% or more total energy derived from protein (Food Safety Authority of Ireland, 2016). In Europe, nowadays, protein intake ranges from 12 to 20% of the total calorie intake (European Food Safety Authority, 2017). Faba bean flour satisfies this criterion and hence it can be considered a high-protein pulse. The protein content in plant-based ingredients is influenced by the variety, geographical region, and grade of processing, which explains the variations between numerous studies [3,10,12]. Most of these studies do not mention the color or size of the faba bean used; hence, future comparative studies based on color and size would be useful in the comparison of nutritional compositions.

### 2.2. Lipids

The fat content of faba bean is relatively low compared to other plant-based protein sources; hence, it can be categorized as a food that is low in fat. The fat content of whole and dehulled faba bean was observed to be 2 g/100 g lower compared to that of buckwheat, lupin, and flaxseed [11]. The fatty acid composition of faba bean oil extracted using hexane contains both saturated and unsaturated fatty acids in equal amounts. The major fatty acids are oleic and palmitic acids. Approximately 56% of the polyunsaturated fatty acids in faba bean are represented by linoleic acid and other fatty acids (elaidic acid 21.2%, hexadecanoic acid 17.6%, α-linolenic acid 4.4%, octadecanoic acid 3.4%, eicosanoic acid 1.3%, and C_20:1_ 0.5%) [33]. The fatty acid composition of faba bean oil extracted by [15] was somewhat comparable, with 97.22% of the total fatty acid composition comprising the following acids: palmitic acid (15.74%), oleic acid (30.12%), linoleic acid (46.41%), stearic acid (2.16%), and linolenic acid (2.76%). Approximately 80% of the total oil is made up of unsaturated fatty acid, resulting in an unsaturated/unsaturated ratio of 4.2. In addition, for the first time, cyclopropanoic acid was confirmed to be present in *Vicia faba* bean in Sudan, which has not been reported in oils from legumes. Analysis of the physicochemical properties of faba bean oil indicated that the iodine value, saponification value, and peroxide value were 114.815 g/100 g, 193.25 mg KOH/g, and 7.64 meq O_2_/kg of oil, respectively. Differential scanning calorimetry suggested the presence of three melting points, −29.14 °C, −4.830 °C, and 31.290 °C, relating to the thermal decomposition of unsaturated and saturated fatty acids in faba bean oil. There is a need for confirmation of these results as there is limited research on faba bean oil, and it is necessary to investigate the presence of other lipids.

### 2.3. Carbohydrates

The main component of faba bean is carbohydrate, with an amount of approximately 63%. Epidemiological studies show that pulses have a low glycemic index in comparison to starchy foods such as potatoes and cereal grains [35]. Hence, the consumption of pulses has been recommended due to their promising effects in preventing and controlling chronic conditions such as cancer, cardiovascular disease, and obesity [12]. Amylose and amylopectin represent the primary components in starch, which significantly affect the functional properties and digestibility of foods. The amylose content of faba-bean-isolated starch was 40%, while the amylopectin content was found to be 60% [36]. In addition, the amylopectin was found in medium-sized granules (degree of polymerization (DP) (13–24)), accounting for 56%, followed by smaller-sized granules (DP 6–12 accounting for 22%), while large-sized granules (DP 25–36 and DP > 36) accounted for 13 and 10%, respectively.

Structural studies of starch isolated from FB using scanning electron microscopy indicated that the starch granules were oval, kidney-shaped, and irregular, similar to other starch granules, with an average size of 10–45 μm. Further analysis by X-ray diffraction revealed that faba bean starch exerted a C-type X-ray pattern. The peak gelatinization temperature and gelatinization enthalpy Δ*H* were found to be 64 °C and 6.5 J/g, respectively [37]. A similar result for faba-bean-isolated starch was also reported by [38]. The Δ*H* of FB starch was comparable to that observed for lentil starch [39] and maize starch [40]. The low gelatinization temperature obtained for faba bean starch may indicate a shorter cooking time.

Whole faba bean has carbohydrate content of 63 g/100 g, of which 39% is mostly insoluble fiber [11]. Mayer Labba et al. [10] reported that the total dietary fiber of cultivars from Sweden varied between 11.337 and 16.59%, among which insoluble dietary fiber was abundant, representing an average of 14%, and with uronic acid and glucomannan making up the majority of all cultivars. However, the dietary fiber content of faba bean showed an average of 12.25–13.49% in a study by Singh et al. [41]. De Angelis et al. [3] examined the proximate compositions of 41 faba bean accessions from 10 different countries. Soluble and insoluble dietary fiber was higher in whole faba bean compared to dehulled faba bean. The insoluble fiber content was observed to be 22.7 g/100 g for whole faba bean, while dehulled faba bean had a lower value of 8.9 g/100 g. A similar trend was observed for soluble dietary fiber, with whole faba bean (2.0 g/100 g) having a higher value than dehulled faba bean (1.3 g/100 g) [11]. Recently, dietary fiber has gained attention as it has been found to provide several health benefits [42]. The insoluble and soluble fiber content of faba bean, at 9.07 and 4.74 g/100 g, was higher when compared to wheat flour (Millar et al., 2019 [43]). The European Food Safety Authority (EFSA, 2010 [44]) recommends daily intake of >25 g of fiber to maintain healthy bowel function, manage weight, and reduce the risk of heart disease and diabetes.

The raffinose family of oligosaccharides (RFO), which consists of raffinose, stachyose, and verbascose, is another class of carbohydrate found in faba bean [45]. Among 15 faba bean varieties that were analyzed, raffinose had concentrations of 1.1–3.9 g/kg, stachyose amounted to 4.4–13.7 g/kg, and verbascose had concentrations of 8–15 g/kg [10]. Landry et al. [45] found a clear correlation between the seed size and RFO content in 40 accessions of FBS. It remains unclear how various food processing activities may affect the presence of raffinose family oligosaccharides, which are implicated in gastrointestinal discomfort, such as gassiness. There is also a paucity of research related to faba bean starch properties; hence, further research in these areas will be useful, as these properties greatly affect the physicochemical properties and functionality. However, different varieties of faba bean seeds may impact different physicochemical and functional properties, and it is necessary to investigate different cultivars. While a few studies have been performed in this context, confirmation by further research is required. These structural properties and physicochemical properties greatly affect faba bean’s application in food systems and its health implications. The starch content and fiber content of faba bean may offer distinctive textural properties, health benefits, and sensory attributes in food systems, but these benefits may be hindered by the presence of RFO.

### 2.4. Minerals

Faba beans represent a reliable source of various mineral elements, such as potassium, calcium, sodium, and magnesium. Whole and dehulled faba bean was observed to be a rich source of K, Ca, P, S, Mg, and trace elements such as Cu, Zn, and Mn [11]. With regard to faba bean flour, the iron and zinc content was significantly higher (131–283% and 142–168%, respectively) than that of wheat flour, although wheat flour demonstrated higher calcium content [32]. Higher mineral content was observed in whole faba bean than dehulled samples [10], indicating the presence of high mineral deposits on the seed coat. However, due to antinutritional substances such as condensed tannins and phytates, the bioavailability of these minerals is often low [46]. The molar ratio of phytates to a certain mineral, such as iron or zinc, is used to determine the bioavailability of that mineral [47]. Since vegetarians have been proven to have an iron shortage compared to non-vegetarians, iron and zinc are the key minerals of concern with a plant-based diet [48]. This requires immediate attention, because plant-based protein sources are becoming increasingly popular, particularly among people who are susceptible to deficiencies, such as children, the elderly, women of reproductive age, and adults. Among 15 faba bean varieties that were analyzed, the molar ratios of Phy:Zn and Phy:Fe in 14 cultivars showed poor bioavailability [10]. Additional research on the impacts of various processing activities on the bioavailability of these minerals is required, as this area has been limited to only iron bioavailability. There has not been any research on the variables that influence the accumulation of these nutrients. Furthermore, it is yet unclear how food processing impacts these minerals’ bioavailability.

## 3. Nutritional Quality

### 3.1. Amino Acid Composition

The nutritional requirements of individuals and animals are not merely based on protein content but specific quantities of essential amino acids. Grain legumes are known to have leucine, lysine, arginine, glutamic acid, and aspartic acid in higher amounts than sulfur-containing amino acids such as cysteine and methionine [49]. The amino acid profile of faba bean isolates is comparable to that of other legumes, with limited sulfur-containing amino acids; these can be supplemented via the incorporation of grains or cereals. In particular, a protein-soluble extract at pH 4 was found to be deficient in tryptophan, isoleucine, and leucine but not in sulfur-containing amino acids. This was due to the presence of albumins, which are soluble at this pH and contain sulfur-containing amino acids [33]. Liu et al. [50] studied the structure and function of faba bean seed storage proteins. There were 497 amino acids in convicilin, and there was a total of three positively charged residues (Cys + Met). Additionally, 46 leucine and 62 glutamic acids represented 12.5% and 9.3%, respectively, of the total amino acids. Legumin A contained 482 amino acids and a total number of positively charged residues (Cys + Met) of 8. With regard to legumin A precursors, arginine was reported as the highest, followed by leucine. Citrate synthase and putative sucrose-binding protein were observed to have the highest (19 and 15, respectively) total numbers of positively charges residues (Cys + Met). The biological value (BV) denotes how much protein would be available for utilization after digestion by an organism. The protein efficiency ratio (PER), which represents the ratio of weight gain and protein consumed by a test group, of protein isolates from alkaline/isoelectric precipitation was found to be higher than 2 (low-quality protein has a value lower than 1.5). This value was calculated using the concentrations of tyrosine, methionine, leucine, and histidine. Furthermore, the theoretical biological value of protein isolates was found to be 47, compared to that of raw flour, which was 40.2 [33].

The amino acid levels of faba bean’s protein-rich fraction (FBC) and isolate (FBI) were similar, with the exception of essential amino acids, where FBI had slightly higher levels than FBR. The amino acid level was above the recommended levels (WHO, 2007), except for sulfur-containing amino acids (SAA), which were low. The sulfur-containing amino acids, as a fraction of the WHO adult requirement, showed amino acid scores of 0.62 and 0.53 for a faba bean concentrate and isolate, respectively [34]. Results obtained by Mattila et al. [11] showed that faba bean contain all the essential amino acids, with a limiting factor of a lower methionine concentration. A comparable finding was reported by Millar et al. [32]. The total amino acid level was observed to be 29.44 and 32.42 in whole faba bean and dehulled faba flour, respectively. Both samples contained all essential amino acids (EAA), including lysine and leucine, which were the most available, while SAA levels were low. The nutritional value of a specific food material can be expressed in terms of Leu and Tyr content (PER value) or other classifications, such as the chemical score for essential AA [11]. The PER value of whole faba bean and dehulled faba bean ranged between 2.53 and 2.43 g/100 g, respectively. In a similar study of Swedish fava bean cultivars, leucine ranged from 50.8 to 72.01% while lysine varied between 44.8 and 74.8 mg/g of protein [31]. Large variations in the content of amino acids were observed between the cultivars, indicating the relevance of cultivar selection based on the nutritional and antinutritional content and intended use. Based on a total protein requirement of 66 g/kg body weight, the EAA are equivalent to those in other high-protein sources and are sufficient for adults, according to the WHO and FAO. When the amino acid composition of whole faba bean is compared to that of other protein products, the impact of the protein content can be seen, as presented in Table 2.

### 3.2. Digestibility

Protein bioavailability in plant-based sources is lower in comparison with animal proteins due to the high content of antinutritional factors (tannins, phytates, and enzyme inhibitors) and poorly digestible protein fractions (Boye et al., 2010a [54]). The digestion of dietary protein from legumes depends on the availability and activity of specific enzymes, as well as the presence of certain compounds. In faba bean, the digestion of protein and starch is mostly affected by antinutrients such as lectins, trypsin inhibitors, and tannins. Luo et al. [26] demonstrated that raw white and green faba bean seeds had 73.28 and 72.65% digestibility, respectively, which is comparable to that of other legumes. However, the processing of faba bean prior to utilization tends to increase its in vitro protein digestibility by reducing the antinutrients. Other characteristics, such as the cell wall rigidity and fiber content, could also play a key role in protein digestibility. Vogelsang-O’Dwyer et al. [34] investigated the in vitro protein digestibility (IVPD) of a faba bean concentrate (FBC) and isolate (FBI). Pepsin digestibility was found to be 5.4–6.4%, whereas the overall protein digestibility values were 22.2 to 26.2% (short-term), 25.1–29.9% (mid-term), and 32.9–39.2% (long-term). Between FBC and FBI, the pepsin digestibility and overall protein digestibility were higher in FBI. This result indicates that the aqueous isolation of protein is useful in improving protein digestibility, which may be attributable to the reduction of antinutrients (e.g., trypsin inhibitors) and smaller amounts of dietary fiber and cell wall interferences. Currently, there is a paucity of information on the digestibility of faba bean concentrates and isolates extracted using different methods. As shown in Table 3, the in vitro protein digestibility can be improved or radically decreased depending on the type of technology used. Research conducted by Sánchez-velázquez et al. [34] on extrusion, cooking, and baking resulted in an improvement in IVPD. A similar improvement in in vitro protein digestibility was observed for irradiation treatment, microwave treatment, annealing treatment, heat moisture treatment, and soaking [26,55]. Table 3 shows the influences of different technological processes on in vitro protein digestibility.

## 4. Bioactivity

Proteins from pulses have been investigated as a source of bioactive peptides [19]. Bioactive peptides are short-chain amino acid sequences released from precursor proteins via enzymatic digestion and can interact and modify specific sites, thereby conferring several physiological benefits beyond normal nutrition (López-Barrios et al., 2014 [64]; Möller et al., 2008 [65]). Bioactive peptides exhibit numerous types of biological activity in vivo and in vitro—for instance, antihypertensive, antimicrobial, anticarcinogenic, hypocholesterol, antithrombotic, mineral-binding, and growth-promoting properties [18,19,66]. Thus, this section reviews the bioactive properties of faba bean, as shown in Figure 3.

Faba bean proteins are a promising source that have been explored for their potential to generate bioactive peptides during the past decade. Faba-bean-derived peptides, controlled hydrolysis, have been studied in various research works and are summarized in Table 3. The inhibition of angiotensin-converting enzyme (ACE); anticarcinogenic, antioxidant, and hypocholestrolemic effects; and antimicrobial activity, tyrosinase inhibitory activity, and serum glucose regulation have been explored in faba bean peptides. Bioactive peptides (BPs) are generated during gastrointestinal digestion; however, in vitro methods employ gastrointestinal enzymes such as trypsin, pepsin, and pancreatin [13,17,67,68,69]. Another equally effective method of peptide production involves germination as a natural hydrolytic process utilizing the action of intrinsic seed enzymes [70].

The authors of [67] subjected FBC to enzymatic hydrolysis in a sequential order, first with trypsin, followed by chymotrypsin and pancreatin. After 3 h of enzymatic exposure, there was an increment in the degree of hydrolysis of 17.1, 9.4, and 14.4% for trypsin, pancreatin, and chymotrypsin, respectively. Among the enzymes used, trypsin showed the highest antioxidant activity in comparison with the other enzymes for the hydrolysates obtained. Mice fed FBH displayed a decrease in atherogenic markers induced by high-density lipoprotein cholesterol (HDL-C), which indicated the presence of bioactive peptides. An interesting observation was that a reduction in atherogenic markers was achieved at a low dose (10 mg/kg). A similar work by Ashraf et al. [71] involved the exposure of FBI to sequential in vitro gastrointestinal digestion using pepsin and trypsin, with and without heat treatment. Hydrolysates produced from heat-treated FBI showed a higher degree of hydrolysis compared to unheated FBI. Size exclusion chromatography of the hydrolysates showed peptide fractions ranging from 500 to 1000 Da, with a high concentration of lower fractions (1–3 kDa). Peptides obtained from the study showed excellent scavenging activity when using the 2,2-diphenyl-1-picrylhydrazyl (DPPH) assay, as well as the potential to reduce Fe^3+^ to Fe^2+^. To evaluate the cholesterol-lowering activity, an in vitro cholesterol micelle model was used. There was a noticeable increase in the inhibition of cholesterol solubilization into micelles, which was attributed to the presence of a high concentration of hydrophobic amino acids and aromatic side chains [13,72].

Hydrolysates are usually purified to identify specific peptides using ultrafiltration and various chromatographic methods, and the characterize peptide sequence is finally characterized using mass spectrometry, ionization techniques, and electrospraying. In silico approaches use computer-aided databases to predict the bioactive peptides in a precursor protein. One commonly used database is BIOPEP, which allows the forecasting of peptides from known amino acid sequences [73]. Karkouch et al. [74] isolated and identified several peptide sequences from FBH using strong cation exchange chromatography, followed by LC-MS/MS with an orbitrap hybrid mass spectrometer. The following seven peptides, designated P1 to P7, were discovered: GGQHQQEEESEEQK (P1), ENIQPAR (P2), IINPEGQEEEEEEEEEK (P3), GPLVHPQSQSQSN (P4), LSPGDVLVIPAGYPVAIK (P5), VESEAGLTETWNPNHPELR (P6), and EEYDEEKEQGEEEIR (P7). Among these peptides, five were found to possess antioxidant activity, with P6 having the strongest radical-scavenging ability. This was ascribed to the presence of aromatic amino acid residues (tryptophan), as well as valine at the N-terminal (Li et al., 2011 [75]). Peptide P5, LSPGDVLVIPAGYPVAIK, exhibited ferrous chelating ability, while P7, P6, and P1 demonstrated the inhibition of *P. aeruginosa* biofilm formation.

Hydrolysates from the same parent protein can generate different peptides depending on the enzyme used, degree of hydrolysis, and enzyme/substrate ratio. Germination also represents an alternative to improve the production of bioactive compounds as it involves enzymatic hydrolysis, which breaks down seed proteins. Bautista-Expósito et al. [54] subjected sprouted faba bean flour to in vitro simulated gastrointestinal digestion. Peptides obtained from germination and subsequent enzymatic hydrolysis were analyzed for their ability to inhibit angiotensin-I-converting enzyme (ACE) and scavenge peroxyl (ORAC and ABTS methods). Their result showed that the germinated FB hydrolysate had high antihypertensive potential (IC50 = 0.63 mg/mL). Sprouted FB showed increased antioxidant activity compared to untreated FB flour. The improved antioxidant activity of the hydrolysate from spouting could be due to the presence of increased peptide availability. However, the presence of polyphenols, vitamins, and free amino acids may also have contributed to the enhanced antioxidant activity. Various peptides were identified after sprouting using UPLC-MS/MS, indicating the role of peptides in radical scavenging and ACE inhibitory properties. The FB peptide LSPGDVLVIPAGYPVAIK shares the amino acid sequence VIPAGYP, which has been identified to possess antioxidant properties [28]. Although ACE inhibitory peptides are generated during germination, variations in temperature and time, as well as the type of legume, can affect the production of these peptides [29]. Further research is needed in this area regarding the use of different enzymes and optimized enzyme/substate concentrations and time, to obtain different peptides with specific bioactivity. Numerous studies of faba bean bioactivity have been performed regarding its anticancer [14,76] and antioxidant activity [13,18,67,68,69,77,78,79], hypocholesterolemic effects [67,77], angiotensin-I-converting enzyme (ACE) inhibition [68,69], metal chelation [74], serum glucose regulation [18], tyrosinase-inhibitory activity [13,80], and antimicrobial effects [13].

## 5. Major Antinutritional Factors

The presence of antinutritional factors in legumes is an adaptation mechanism to protect them from adverse environmental conditions; however, these antinutrients limit the utilization of legumes in food and have been shown to have health consequences. The most limiting antinutrients in *V. faba* are the pyrimidine glucosides vicine and convicine [81]. As indicated by Amarakoon et al. [82], a suitable ratio of antinutrients to nutrients can minimize the negative effects of antinutrients on digestibility, while playing a significant role in cellular processes such as anti-inflammatory and antioxidant activity. 

### 5.1. Vicine and Convicine

Despite the numerous advantages of faba bean seeds, their production and utilization have historically been constrained because they contain the pyrimidine glycosides vicine and convicine, which are present in roughly 1% of the dry matter in the cotyledons of most faba bean varieties [82]. Different varieties of faba bean have been bred containing one tenth of the vicine and convicine content; however, a reduction to zero has not been attained yet. Various researchers have shown that late-harvested seeds tend to have lower levels of pyrimidine glycoside content compared to early-harvested seeds [32,45]. The degradation of β-glycosidic linkages leads to the transformation of vicine and convicine into their corresponding aglycones, respectively, divicine (2,6-diamino-4,5-hydroxypyramidine) and isouramil (6-amino-2,4,5-trihydroxypyramidine) (Figure 4). Hydrolysis occurs either through enzymatic action (β-glucosidase) during seed germination or by microbial action digestion in the large intestine [29]. These generated aglycones lead to a condition called favism, characterized by hemolytic anemia [21,83]. This condition is prevalent in individuals with a deficiency in glucose-6-phosphate dehydrogenase (G6PD). G6PD’s function is to defend against oxidative stress in cells by triggering reduced nicotinamide adenine dinucleotide and replenishing reduced glutathione; hence, a reduction in their activity leads to oxidative stress, resulting in hemolytic anemia [29]. The vicine and convicine content was reported by Mayer Labba et al. [10] for different cultivars of fava bean in Sweden. In general, among all the cultivars, the vicine and convicine content ranged from 403 to 7014 μg/g. In a study by Khazaei et al. [84], the Melodie cultivar was found to contain vicine and convicine at lower levels of 90 and 14 μg/g, respectively.

Vicine and convicine are thermostable; however, their concentrations can be lowered substantially using different processing methods. Processing techniques such as soaking, roasting, boiling, microwave, fermentation, irradiation, and frying can reduce the content of vicine and convicine in faba bean [29,85,86]. In addition, alkaline extraction followed by isoelectric precipitation can also reduce the content of vicine and convicine; however, this method may be costly and require large amounts of energy. FBPI showed a ratio of vicine to protein of approximately 0.034 to 100 *w*/*w*, indicating 96–99% lower vicine content [87] compared to the ratio of vicine to protein in whole faba bean [88,89]. The method of production of FBPI caused a substantial reduction (96–99%) in convicine content: in each step of the extraction process, the aqueous medium dissolved alkaloids and hence they could further be separated from the protein following centrifugation. Currently, breeding has been targeted as an approach to reduce the content of vicine and convicine and this could represent the optimal solution.

### 5.2. Phytic Acid

Phytic acid (inositol dihydrogen phosphate (IP6) or inositol polyphosphate) is a common antinutrient present in FBS and is stored in the form of phosphorus in dry beans. The average phytic acid content in faba bean is approximately 8.58 mg/g [90]. At physiological pH, polyphosphate is partially ionized and transformed into phytate anion. Phytate is a cheating agent that binds divalent cations such as iron, thereby reducing their bioavailability. From a nutritional perspective, in dry beans, phytic acid is implicated in their ability to reduce the bioavailability of essential minerals and possibly interferes with protein utilization through phytate–protein or phytate–protein–mineral complexes [90]. Wet processing such as germination, fermentation, and soaking has been shown to minimize the phytic acid content and enhance the solubility of iron in food, thereby improving the bioavailability of minerals in legumes [91]. According to a study [10], the phytate value of faba bean, depending on the cultivar, varied between 112 and 1281 mg/100 g, which indicates large variation among cultivars. Faba bean from China was reported to have phytate content of 823 mg/100 g [92]. Other researchers have reported comparable results for Bolivian fava bean (1170 mg/100 g) [93]. These values are, however, lower compared to that reported for soy (3170–3900 mg/100 g) [65].

To guarantee the appropriate bioavailability of zinc, the European Food Safety Authority (2017) advises increasing zinc intake in the case of high-phytate meals (300–1200 mg/day) by 2–7 mg/day. Differences in phytate content have been attributed to factors such as location and climatic conditions. The total phytate concentration among 41 faba bean accessions ranged from 13 to 16 mg/g, with outliers with a high concentration of approximately 23 mg/g dry matter. This indicates the importance of screening for physiochemical, antinutrient, and functional properties in selecting faba bean cultivars with desirable attributes for food applications [3].

### 5.3. Condensed Tannins

Tannins are secondary metabolites of varying chemical structures produced in plants. They are largely classified into hydrolysable tannins and condensed tannins [68]. Tannins function as multidentate ligands that promote protein crosslinking; hence, high-molecular-weight proanthocyanidins can precipitate proteins easily [94,95], while ellagitannins with rigid conformation tend to precipitate proteins less effectively [96]. The biological relevance of tannins can be considered in terms of different aspects; they are non-absorbable compounds that generate local health benefits in the gastrointestinal tract through radical scavenging and antiviral, antimutagenic, and antinutritional effects during their colonic fermentation, which lead to health implications in several organs [68]. The content of tannins is reported to range from 4 to 6 mg/g in various faba bean cultivars [26,97]. Different processing methods, such as dehulling, soaking, cooking, autoclaving, fermentation, and extrusion, have proven to be successful in eliminating or reducing the levels of tannins in FBS [26,36].

### 5.4. Oligosaccharides

Low-molecular-weight carbohydrates having α-galactosidic and β-fructosidic linkages are known as oligosaccharides. Raffinose-series oligosaccharides (raffinose, verbascose, and stachyose) and sucrose have been implicated in digestive discomfort, such as gas production and flatulence, due to the anaerobic fermentation of these carbohydrates [75]. Various concentrations of raffinose-family oligosaccharides (RFO) were observed in 15 cultivars of FBS, in the range of 8–15 for verbascose, 1.1–3.9 g/kg for raffinose, and 4.4–13.7 for stachyose [10]. Similar studies on 40 cultivars of faba bean also showed variations in RFO content, with verbascose content of 6.7–50.3 g/kg, 2.7–13.0 g/kg raffinose content, and 9.0–25.0 g/kg stachyose content [45]. RFO have adverse health effects and they have been implicated in flatulence and discomfort due to anaerobic fermentation by microorganisms in the digestive tract [98]. Meanwhile, low levels of RFO have been shown to be beneficial to human health. The health benefit of RFO is observed in their prebiotic nature, increasing the bifidobacterial population in the gut microbiome, promoting the absorption of minerals, and enhancing the immune system while protecting against colon cancer [99]. It is important to make a well-informed decision, taking into consideration various components that are useful in selecting faba bean cultivars in order to limit the presence of antinutrients and optimize the nutrients. In a study by Landry et al. [45], a modified HPLC method was used to quantify low-molecular-weight carbohydrates. The RFO and sucrose content was high in mature seeds, while sucrose was predominately found in immature seeds (13.4%). Verbascose (2.4%) and stachyose (1.9%) were the main RFO observed in mature seeds across all populations. 

### 5.5. Other Antinutritional Factors

Other antinutrients present in faba bean include trypsin inhibitors, which have a crystalline globulin structure and retard growth in mammals and chickens by causing pancreatic hypertrophy [100]. It has been shown that the presence of trypsin inhibitors during digestion inhibits the activity of enzymes such as trypsin, chymotrypsin, and pancreatic enzymes, hence reducing the digestion and absorption of protein. Thus, it is necessary to process plant products containing low trypsin inhibitor content [101]. The trypsin inhibitor content varied from 1.2 to 23.1 trypsin inhibitor unit per mg for 15 different varieties of faba bean [10]. Millar et al. [32] and Shi et al. [23] reported trypsin inhibitor content of 5.45 and 5.96, respectively, for faba bean. However, for soybean, a TIU/mg of 45.89 was reported, which is higher than that observed in faba bean. Another antinutritional factor of relevance is lectins. A report on the antinutritional content of various cultivars of faba bean showed lectin content ranging from 0.8 to 3.2 HU/mg using human erythrocytes [10]. Canadian faba bean has been observed to contain a lectin concentration of 5.52 HU/mg [24], while Spanish cultivars showed lectin content of 49.3 HU/mg, which are both relatively high, with both researchers using rabbit red blood cells. It is worth noting that the hemagglutination assay, which is mostly employed to estimate the concentrations of lectins in faba bean, is a semi-quantitative approach based on the agglutination of phytohemagglutinin to erythrocytes [102]; hence, huge variations in the lectin content of faba bean have been reported.

## 6. Processing Effect on Toxicants and Pharmacological Activity

Thermal processing can enhance the digestibility of proteins via the denaturation and unfolding of protein structures, as well as inactivating certain antinutrients; however, certain methods of processing can negatively impact digestibility due to extensive aggregation [103]. Several processing techniques, such as soaking, cooking, dehulling, and roasting, are employed to obtain improvements in the nutritional profiles of pulses through the reduction or elimination of antinutritional factors [104]. It has been established that non-thermal food processing techniques as shown in Table 4 provide the lowest risk to food safety and could serve as an alternative to thermal processes. Since these processes also affect cell permeability and enhance bioactive properties, they have also been explored as emerging treatments for the processing of different food materials [105,106].

### 6.1. Soaking

Faba bean seeds are usually subjected to pre-treatment methods such as soaking prior to further processing, to reduce the antinutrient content and cooking time. The amount of phytic acid in faba bean was significantly reduced after soaking, dropping from 21.1 g/kg to 14.6 g/k, and it reduced the condensed tannin content compared to raw faba bean (1.02 vs. 1.95 g/kg). This could be attributed to the leaching of compounds from the interior of the seed into the liquid media, which is beneficial in lowering the levels of antinutrients. Alonso et al. [107] suggested that steeping faba bean seeds prior to processing can reduce the concentrations of certain antinutrients. The soaking of faba bean for 4 h at room temperature resulted in a reduction in hemagglutination activity, oxalate content, and phytic acid content [23]. The modulation of soaking using an acidic or alkaline component has been shown to cause a 100% reduction in the levels of vicine and convicine [108], primarily due to the hydrolysis of these pyrimidine glycosides at an acidic or basic pH. 

Setia et al. [109] reported that the soaking followed by the germination of faba bean seeds for up to 72 h did not result in any significant alteration in the chemical constituents of the flour produced, despite an improvement in α-amylase activity; however, significant changes were observed in the pasting, emulsifying, and foaming properties after germination, which was attributed to partial protein denaturation and the dissociation of proteins, which improved the surface activity. Soaking faba bean in water at 30 °C for 12 h has been demonstrated to cause a 14.9% decrease in α-amylase activity. Lower levels of α-amylase inhibitors in soaked beans have been ascribed to the seeping of compounds during soaking [110]. Steeping caused a significant decrease in the percentage of trypsin inhibitors (12.73 vs. 22.59%) and chymotrypsin inhibitors (11.43 vs. 17.51%) in faba bean [104]. Hemagglutination activity was also reduced to approximately 0.62–5.18% following soaking for 4 h at room temperature [24]. Unfortunately, steeping may lead to the partial loss of soluble proteins during the leaching of antinutrients such phytates and oligosaccharides. The authors of [111] showed that there was no correlation with the total water absorbed and protein content in faba bean. Additionally, the temperature and the strength of the solution had a positive correlation with how quickly the seeds absorbed water. The temperature of soaking may affect the hydration rate of faba bean. Thus, research on the hydration kinetics of faba bean needs to be performed, as well as considering the effect of the soaking temperature on the nutritional and physicochemical properties. In addition, research on the impact of ultrasound-assisted steeping and other technologies to reduce the soaking time of faba bean is yet to be documented. Moreover, the discarded water after soaking may contain valuable compounds that could be exploited.

The soaking process may exert additional benefits by modifying the process through the addition of exogenous compounds such as Ca^2+^, Mg^2+^, Na^+^, L-glutamic acid, L-glutamate, or other phytochemicals as liquid media to favor or direct the metabolic pathway. This is very useful if the antinutrient content, nutritional profile, or secondary metabolites need to be altered. 

### 6.2. Dehulling

Dehulling is a process in which the hulls are detached from the cotyledons of pulses [112]. The hulls from seeds can then be utilized in the production of phytochemicals, while the cotyledon serves as a rich source of plant protein [113]. The protein content of faba bean before dehulling varied from 30.1 to 34.8%. After dehulling, the protein content was found to range between 60.0 and 60.9%. The seed coat from faba bean contained residual protein, since small quantities of broken kernels were present in the hull [85]. Similar studies on dehulling showed increased protein content from 27 to 31.3% [107]. Because tannins are mostly found in the seed’s testa, the dehulling of faba bean seeds showed significant reductions in tannin content (0.15 vs. 1.95 g/kg) [107]. Hulling has resulted, in some cases, in increased trypsin inhibitory activity, since trypsin inhibitors are found in the cotyledon fraction [43,114]. The dehulling of faba bean was also shown to increase the levels of phytic acid, with smaller amounts of phytates in the hulls [115]. This is valuable when considering the implications of hulling in relation to the specific nutritional quality of the intended ingredients, as iron absorption is inhibited in the presence of phytic acid. In addition, the dehulling of seeds has also been linked to improvements in the palatability and taste of some legumes [116], and this needs to be investigated in faba bean seeds.

### 6.3. Germination

Germination treatment has been used in various applications to minimize cooking difficulties, such as reducing the cooking duration, enhancing the nutritional quality, eliminating and/or reducing the levels of antinutrients [117], and improving the flavor characteristics [118], and it is an effective means for the diversification of the techno-functional properties [109]. As protein and carbohydrate represent the key sources of energy during germination, various biochemical and structural changes occur, which affect the nutrient availability. Storage proteins undergo hydrolysis during germination, thus increasing the amino acid content in pulses. As with other grains and pulses, faba bean germination involves soaking and disinfection followed by germination [119].

Various research works have shown germination to be an efficient method in reducing the phytic acid content in grains and legumes [120,121] due to the increased phytase activity during germination. Germination at different time intervals resulted in a drastic decrease in phytate content after 12, 24, and 72 h of germination, with the longer period of germination being more effective in reducing the levels of phytic acid from 21.1 to 8.5 g/kg [107]. While there have been numerous reports on improved mineral content, such as iron and calcium content, during germination [122], there is a poor correlation with improved absorption rates for these minerals, mainly due to the influence of other secondary cations that may be present and hence potentiate the precipitation of minerals in the presence of phytates. Intensive research is needed in this area to monitor the impact of germination on mineral absorption in the presence of secondary cations such as magnesium. Alonso et al. [107] showed that after germination for 24, 48, and 72 h, the condensed tannin content was reduced significantly to 0.86, 0.82, and 0.78 g/kg, respectively, compared to raw faba bean (1.95 g/kg), and they also noted improved protein digestibility due to the reduction in the levels of antinutrients. After processing, trypsin inhibitors and other antinutrients may be present at lower quantities, which may explain the improvements in protein digestibility. Research by Bautista-Expósito et al. [54] also indicated the effectiveness of germination in reducing the levels of phytic acid (~50%), condensed tannins (~35%), and trypsin inhibitors (~35%).

Germination therefore represents one of the best non-thermal food processing techniques in improving the general physiochemical, nutritional, metabolite composition, and structural properties of faba bean because of the low processing temperature, safety during processing, and sustainability of the method. Additionally, germinated seeds could be modified by ultrasound or irradiation to ensure a greater improvement in functional components such as phenolics or to reduce the levels of antinutrients. To conclude, germination lone or in combination with other processes is a cost-effective, environmentally sustainable technique for the rapid functionalization of faba bean seeds.

**Table 4 molecules-28-05431-t004:** Non-thermal technologies to reduce and eliminate antinutrients in faba bean seeds. Symbol ↑ donate improvement and ↓ indicate reduction.

Non-Thermal Technology	Processing Conditions	% Major Toxin Reduction	Reference
Extraction	Protein concentrate Protein isolate	↑Trypsin inhibitors increased by 64.79% ↓Vicine reduced by 8.5% ↓Convicine reduced by 12% ↓Trypsin inhibitors reduced by 80% ↓Vicine 100% reduction ↓Convicine 100% reduction	[34]
	Alkaline extraction (pH 10.5); protein isolate	↓Vicine and convicine reduced by 99%	[33]
Germination	Time: 0–9 days; temperature: 20 °C	↓Vicine reduced by 0.7–9.8%	[88]
Germination	Germination at 24–72 h at 25 °C	↓Phytic acid content reduced by 53.46–61% ↓Condensed tannins reduced by 56–60%	[43]
Germination	Time: 0–144 h	↓Trypsin inhibitors (TIU/mg) reduced by 26–38% ↓Phytic acid (mg/g) reduced by 55–58% ↓Condensed tannins reduced by 29–39%	[54]
Fermentation	Fermentation with Lactobacillus plantarum (DPPMAB24W); time: 0–48 h; temperature: 30 °C	↓Convicine content reduced by 14–93% ↓Vicine content reduced by 35–96%	[29]
Fermentation using *L. plantarum*	Fermentation with *L. plantarum* for 24 h at 25 °C	↓Phytic acid (mg/g) reduced by 8%	[123]
Fermentation	Fermentation with starter culture; time: 1–3 days at 30 °C	↓Phytic acid (mg/g) reduced 48–84%	[22]
	*L. plantarum* (sourdough) 48 h at 30 °C	↓Vicine and convicine reduced by 100%	[56]
Enzyme treatment	Flour treated with phytase (0, 2, 10, and 20 U) for 1–24 h using *L. plantarum*	↓Phytic acid (mg/g) reduced by 15.46–90.72%	[123]
Dry fractionation	Micronized flour, protein-rich fraction, coarse-starch-rich fraction, wet-extracted protein fraction, deproteinated starch-rich flour	Phytic acid (mg/g) content was 4.3, 10.5, 7, 10.2, and 1, respectively Trypsin inhibitor (TIU/g) content was 10.5, 14.5, 5.8, 14.8, and 3.2, respectively	[124]
High-pressure processing	Concentrate (600 MPa for 4 min)	↓Relative trypsin activity was reduced by 90.8%	[125]

### 6.4. Thermal Processing

The thermal processing of food materials can occasionally be inefficient, altering the organoleptic properties and lowering the nutritional quality. It is crucial to understand that there are several levels of heating-based treatment. Thermal processing can provide mild or severe heat treatment depending on the amount of heat applied and the application time. Because of their ability to guarantee microbial stability, ensure product stability, and deactivate spoilage enzymes, the use of thermal processing techniques remains the most common method in the food industry [21]. Efforts to improve the application of FB have utilized a wide range of thermal processing methods, such as cooking, roasting, extrusion cooking, and microwave-assisted cooking, to reduce the unpleasant “beany flavor” and antinutritional factors [126].

Cooking of faba bean showed approximately 93.77–99.81% elimination of hemagglutination activity [24]. After cooking faba bean at 95 °C for 1 h, the complete elimination of chymotrypsin inhibitor activity was observed due to their high sensitivity to heat [104], indicating the potential of thermal treatment in the elimination of antinutritional factors. The reduction in the total oxalate content of cooked, pre-soaked faba bean was also found to be approximately 30% [24]. Convectional heating with hot water heats the plant tissue interior during cooking. The cell walls, membranes, and cell organelles undergo modifications as a result of the thermal treatment, affecting their structures. Mostly, these modifications cause the leaching of phytochemicals into the liquid media, resulting in either a positive (e.g., loss of toxins) or negative effect (leakage of valuable compounds). The amount of compounds lost mostly depends on the cooking temperature and duration [127]. The vicine and convicine content can also be greatly impacted by thermal treatment, since these proteinaceous compounds are susceptible to denaturation and degradation. Heating of faba bean at 122 °C resulted in a vicine reduction of 30% and a convicine reduction of 61%. 

Microwave heating could be used to improve the flavor profile of untreated faba bean seeds, as evidenced by the inactivation of endogenous peroxidase and lipoxygenase, which are responsible for the beany flavor in faba bean [128]. According to Kaderides et al. [129], the volumetric thermal effect underlies the majority of microwave heating, which causes cell leakage and an increased mass transfer rate. This process shortens the extraction time by quickly and effectively producing heat in the volume of the material. Microwave cooking of faba bean seeds resulted in a considerable reduction in the levels of tannins and tryspsin inhibitors [26]. An improvement in protein digestibility has been achieved using cooking and autoclaving due to the elimination or reduction of antinutrients [26]. Generally, antinutrients such as condensed tannins and polyphenols bind proteins to form complexes, thereby impairing proteases’ access to peptide bonds [130]. Thermal treatment tends to minimize these interactions and additionally cause structural changes in the protein that could improve protease accessibility and improve functionality. Another thermal processing method commonly used in pulse treatment is extrusion. Extrusion has been shown to have no impact on protein content; however, there was an observed reduction in phytic acid content after extrusion cooking (15.9 g/kg), which was significantly high compared to raw faba bean (21.1 g/kg) [107]. High-performance liquid chromatography was used to reveal that, during extrusion processing, compounds such as inositol hexaphosphate were hydrolyzed into penta-, tetra-, and triphosphates [68]. Extrusion cooking was also effective in reducing the condensed tannin content compared to raw faba bean (0.89 vs. 1.95 g/kg), as well as improving in vitro protein digestibility (87.1 vs. 70.8%) [107]. The relationship between different thermal processing methods and cell wall rigidity and dietary fiber may provide information that is useful in improving the technological properties, but it needs further attention as there is little information on this area in faba bean. Based on these findings, it is crucial to optimize the thermal processing conditions as shown in Table 5 in such a way that preserves the product’s sensory, nutritional, and pharmacological qualities while also extending its shelf life.

## 7. Impact of Processing Technologies on Faba Bean Seeds’ Pharmacological Activity

Natural substances with phenolic structures fall within the category of polyphenols. There are four main subclasses in this family: lignans, stilbenes, phenolic acids, and flavonoids. Polyphenols have free-radical properties; preventive effects against cancer, cardiovascular disease, and other age-related disorders; and the ability to reduce inflammation and allergies [133,134]. Angina pectoris, cervical lesions, chronic venous insufficiency, dermatopathy, diabetes, gastrointestinal illnesses, lymphocytic leukemia, menopausal symptoms, rhinitis, traumatic cerebral infarction, etc., have also been observed to benefit from flavonoids [135]. 

Variations in polyphenol content in various parts and varieties of faba bean have been reported. The authors of [10] showed that the phenolic content of faba bean cultivars varied between 1.4 and 5.0 mg/g, with colored varieties accumulating the highest phenolic content. A previous study showed that the total phenolic content of 41 faba bean varieties was observed to range from 2.5 to 3.2 mg/g, with two outliers having shown higher concentrations of 4.5–5.3 mg/g. The antioxidant activity of all 41 faba bean samples showed an average of 5 μmol/g [3]. The total phenolic compounds in raw faba bean from India were also observed amount to 0.45 mg/g [136]. The concentration of total phenolics was found to be 387.52 mg/100 g in faba bean flour, while that of wheat flour was 68.27 mg/100 g [32]. Phenolic acids and tannins both contribute to the total phenolic acids, which have a significant effect on the digestibility of other nutrients (Sarwar Gilani et al., 2012 [137]). The authors of [138] profiled the phenolic content of ten Australian faba bean cultivars. The TPC content of the 10 accensions varied between 258 and 570 mg/100 g.

An analysis of the phenolic profile using high-performance liquid chromatography coupled with an ultraviolet detector (HPLC-UV) revealed that the most abundant flavonoids were catechin and rutin, whereas, in the case of phenolics, syringic acid was the most abundant (72.4–122.5 mg/kg). However, a low concentration was found for chlorogenic, vanillic, protocatechuic, p-coumaric, p-hydroxybenzoic, and trans-ferulic acid. The results showed that the phenolic content varied significantly among the cultivars, while, in some cases, the growing site showed a negligible impact. In the study by Valente et al. [139,140], the phenolic composition and antioxidant activity of European FB cultivars varied significantly between cultivars. A study by Baginsky et al. [73] also reported that differences existed in the phenolic profiles among ten immature Chilean faba bean varieties. A comparison of the TPC and flavonoids of 13 faba bean varieties analyzed at various stages of maturity revealed variations in total phenolic content and flavonoid content, with higher levels observed at the maturity stage [141]. A comparative data analysis of these results is recommended as the extraction method and analytical method could play a key role in the differences in phenolic content. Specific polyphenols can bind iron, thereby inhibiting iron bioavailability during the consumption of faba bean [142]. Moreover, phenols have been described to have radical-scavenging properties, with significant health benefits [143]. Polyphenols are known to cause oxidation, resulting in the discoloration of flour and its protein preparations [130]. This issue may be mitigated with the addition of sodium sulfite or the extraction of polyphenols during protein extraction, which can then be used in other applications. Faba beans of various colors must be taken into account in comparison studies, to understand the relationship between the seed color and phenolic profile. 

The need to maintain polyphenols’ nutritional content and enhance their biological function has increased with growing knowledge of the health advantages linked to polyphenols. In some circumstances, different thermal processing pathways boosted the polyphenol content, whereas, in other cases, they lowered it. However, under extreme circumstances, heat processing may result in a number of physicochemical changes that may have a detrimental effect on the sensorial qualities, degrade thermolabile food-borne compounds [21], and possibly generate toxic compounds (such as advanced glycation end products from heat-induced Maillard reactions) [21]. As presented in Table 6, it can be observed that, depending on the processing technique and conditions used, the phenolic content can increase or decrease. Soaking in either cold or hot water, with or without sodium carbonate, has been used to increase the phenolic content of faba bean [144]. After germination for 3 and 5 days, Di Stefano et al. [75] observed an increase in phenolic content. A critical characteristic of germination is the bioconversion and biosynthesis of certain compounds that positively influence human health. Different thermal treatments using extrusion, cooking, and baking resulted in a drastic reduction in antioxidant activity, as measured using DPPH and ORAC [34]. Martini et al. demonstrated that boiling, grilling, and baking had detrimental impacts on the primary polyphenols and hydroxycinnamic acids in aubergine, whereas frying dramatically boosted these compounds due to both isomerization and hydrolysis events [145]. These findings demonstrated that thermal processing could alter polyphenols’ chemical characteristics by epimerization, oxidative polymerization, and degradation. In addition to these reactions, polyphenols could also form complexes with proteins or polysaccharides during thermal treatment. It is possible to increase polyphenols’ resistance to heat destruction by attaching them to proteins. The treatment of a faba bean protein concentrate with the alcalase enzyme resulted in an increment in antioxidant properties of 2–36% [146]. Similar significant improvements have been observed by Samaei et al. after treating faba bean protein with different enzymes [147].

This clearly shows that different processing techniques can have a negative or positive impact on faba bean’s pharmacological activity. It is important to note that non-thermal processing methods that tend to increase the phenolic content and enhance antioxidants may have a drawback, since the enzymes and antinutrients may not be reduced. There is still limited information on the impacts of different processing techniques, such as infrared drying, steaming, roasting, combined treatment, and fermentation with different strains, on the phenolic profile of faba bean, and this area needs to be investigated. Overall, polyphenols can undergo alterations in their composition as a result of thermal processing, including direct interactions mediated by polyphenols with thermal processing intermediates, which may have favorable or negative effects on food processing. 

## 8. Conclusions

The consumer interest in plant-based nutrition has steadily grown in recent years, and organizations, experts, and consumers alike are becoming more concerned about the nutritional and health benefits of sustainable diets. Faba bean seeds are an emerging source of protein-rich material, as well as other macronutrients, such as dietary fiber, starch, and fatty acids. Faba bean is an excellent source of leucine compared to other cereals. Faba bean also has relatively low lipid content, making it a low-fat food. Faba bean seeds also exhibit antioxidant, anti-inflammatory, antimicrobial, serum glucose regulation, and other biological qualities. Antinutritional factors such as pyrimidine glycosides, tannins, and phytates can prevent nutrients from being fully absorbed. The review above has shown that different processing methods, such as germination, soaking, irradiation, fermentation, microwaving, and enzymatic treatments, are utilized to reduce the antinutritional factors and enrich the bioactive components. Numerous studies have shown that faba bean ingredients can be successfully incorporated into the processing of a broad variety of foods, especially to improve the desired nutritional and functional properties of these products. 

## 9. Future Perspectives

Much is still unclear about the chemical makeup, nutritional value, health advantages, processing characteristics, and functional behavior of faba bean constituents in food processing, despite recent advances in understanding. For instance, the starch components and their physicochemical properties have not been thoroughly studied. Different physically, chemically, and enzymatically modified faba bean starches for value-added processing should be investigated further in the future. Furthermore, combining starches from faba bean with other sources of starch may improve their physicochemical qualities and expand their application. Future studies are also required to validate health claims using animal and clinical trials. Finally, the compositional profiles of hulls from faba bean are rarely documented, but this by-product may represent a valuable source of bioactive compounds such as polyphenols and hence needs to be explored. Because faba bean is still in the early phases of development, more research on the combined influence of non-thermal and future thermal treatments on them is essential. A few barriers must be addressed for non-thermal methods to be more efficient, such as undesired color development and nutrient loss into the liquid phase. Finally, various techniques, such as pulsed electric fields, UV light, ohmic heating, cold-plasma technology, vacuum impregnation, ultrasonication, ionization radiation, and microwave treatment, could be used to improve the functionality, reduce the antinutrients, and enhance the pharmacological properties. 

## Figures and Tables

**Figure 3 molecules-28-05431-f003:**
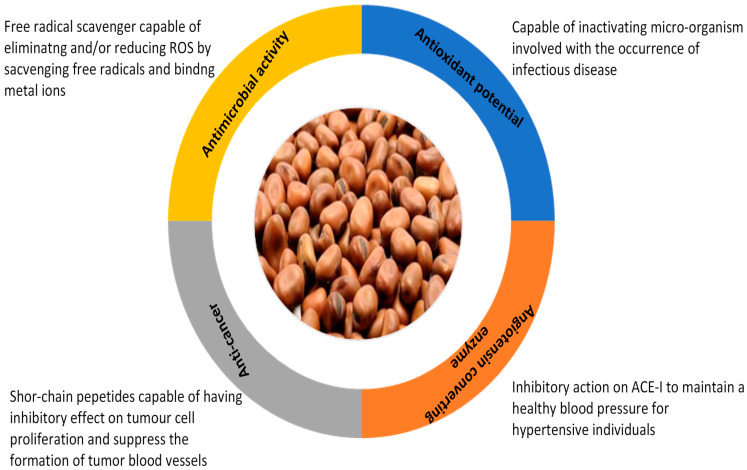
Different health benefits associated with faba bean bioactive peptides.

**Figure 4 molecules-28-05431-f004:**
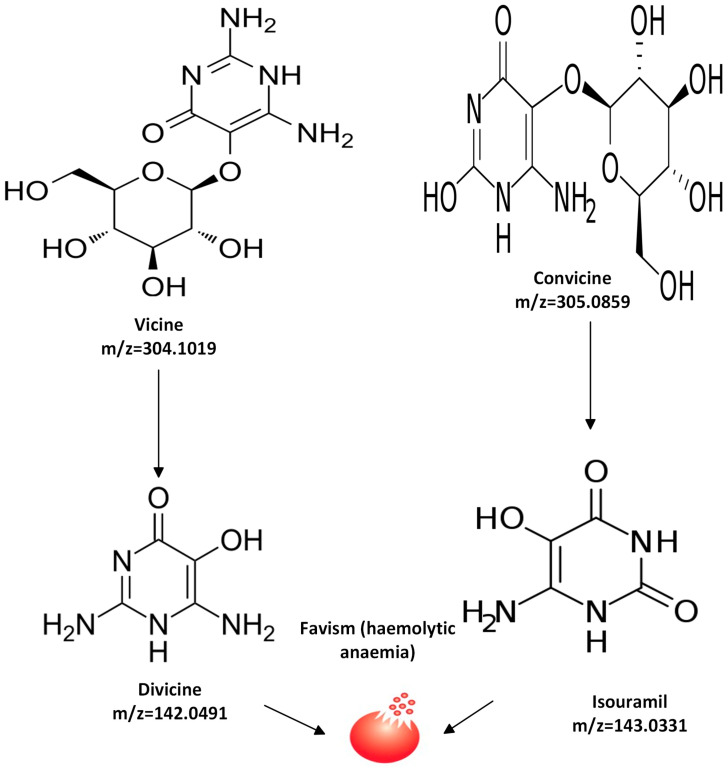
Skeletal structure of vicine and convicine in faba bean (adapted from [20]).

**Table 1 molecules-28-05431-t001:** Proximate compositions of faba bean ingredients. Data sources (^a^ [11]; ^b^ [34]).

Sample	Protein %	Fat %	Ash %	Carbohydrate %
Whole ^a^	30.8–31.60	2–2.2	3.3–3.5	62.5–63.8
Hulled ^a^	35.5	2.1	4.0	57.5–59.1
Protein concentrate ^b^	64.1	2.43	4.8	28.7
Protein isolate ^b^	90.1	4.36	5.2	0.34

**Table 2 molecules-28-05431-t002:** Amino acid profiles (% *w*/*w*) of faba bean flour and protein and selected non-faba-bean products for comparison.

Amino Acids	Faba Bean		Protein Concentrate	Protein Isolate	Other Protein	Casein ^e^	FAO/WHO Requirement	
	Whole Seeds ^a^	Dehulled Seeds ^a^	FBC ^b^	FBI ^c^	SPI ^d^		2–5-Year-Old ^f^	Adult ^f^
Histidine	2.56	2.43	2.39	2.80	2.81	2.70	1.90	1.60
Isoleucine	4.1	3.97	3.73	3.80	4.35	4.90	2.80	1.3
Leucine	7.5	7.25	7.10	8.0	6.79	8.40	6.60	1.90
Lysine	6.43	6.16	6.34	7.0	5.23	7.10	5.80	1.60
Methionine	0.89	0.79	0.60	0.100	0.92	2.60	-	-
Phenylalanine	4.25	4.07	4.13	4.90	5.14	4.50	-	-
Threonine	3.65	3.45	3.54	3.70	3.98	3.70	3.40	0.90
Valine	4.75	4.56	4.14	4.10	4.28	6.0	3.50	1.30
Alanine	3.97	3.78	3.85	4.40	3.72	2.7	-	-
Arginine	9.73	10.21	10.48	10.00	7.35	3.3	-	-
Aspartic acid	11.2	10.71	10.30	13.30	11.47	6.3	-	-
Cysteine	1.18	1.13	-	5.00	0.05	0.04	-	
Glutamic acid	16.78	16.05	16.25	19.90	20.67	19.0	-	-
Glycine	4.38	4.06	3.81	4.90	3.74	1.60	-	-
Serine	4.98	4.74	4.87	6.30	5.32	4.60	-	-
Tyrosine	3.67	3.50	3.05	2.63	3.61	5.50		-
Proline	4.22	4.09	4.24	3.40	5.13	-	-	-

Note: tryptophan was not quantified due to analytical challenges and low quantities. Data obtained from ^a^ [11]; ^b^ [51]; ^c^ [33]; ^d^ [52]; ^e^ [53]; ^f^ [50].

**Table 3 molecules-28-05431-t003:** Processing influence on in vitro protein digestibility (IVPD). Improvement represented by ↑ and reduction by ↓.

Technology	Conditions	Results	Reference
Baking	Bread with 30% faba bean flour; faba bean sourdough bread	↑IVPD: 0.79–16.51 (%)	[56]
Extrusion	Temperature: (140–180 °C); moisture: 18–22%); unsoaked beans	↑IVPD: 3.71–4.26 (%)	[57]
	Temperature: (140–180 °C); moisture: 18–22%); soaked for 16 h at 30 °C	↑IVPD: 2.71–5.80 (%).	
Extraction	Cooked starch extract	↑Rapidly digestible starch increased by 475% ↓Slowly digestible starch reduced by 93% ↓Resistant starch reduced by 87%	[37]
	Protein concentrate Protein isolate	IVPD: 33.9 IVPD: 9.2	[34]
	Protein isolate Optimized ultrasound-treated isolate	IVPD: 68.42 IVPD: 65.98	[58]
Thermal treatment	Extruded flour; cooking flour; baked flour	↓IVPD 2–4(%)	[34]
Soaking	Soaked flour in acidic and acidic + hydrogen peroxide Soaked flour in water and neutral conditions	↑IVPD 11.78–14.75% ↓IVPD 2.0–7.4%	[59]
	Time: 12–48 h at room temperature	↑IVPD 0.82–1.4%	[26]
Thermal treatment	Extrusion (barrel temperature between 30 and 120 °C); cooking; extrusion followed by baking for 27–35 min	IVPD (%) was found to be 82.22, 81.41, and 76.79, respectively	[60]
Cooking	Soaked for 8 h followed by cooking	↑Albumin digestibility 29% ↑Glutelin digestibility 23.61%	[7]
	Cooking (100 °C; 30 min); soaking + cooking (100 °C; 30 min); hulling + soaking + cooking (100 °C; 30 min)	↑IVPD 7.33–9.46 (%)	[26]
Heat treatment	Annealing with incubation at 65 °C for 24 h Heat–moisture treatment; incubation at 120 °C for 24 h	↑IVPD 8.13 (%) ↓IVPD 4%	[55]
Fermented product	Fermented and unfermented pasta formulation (10–50%).	↑IVPD 16.86–81%	[61]
	Bread with 50% faba flour Bread with 50% fermented four using *L. plantarum*	IVPD 53.9% IVPD 72.3%	[62]
Dehulling	Hulled seeds	↑IVPD 0.79%	[26]
Microwave Treatment	Microwave for 6 min; soaking + microwave (6 min); hulling + soaking + microwave (6 min)	↑IVPD 1.75–3.5%	[26]
Autoclaving	Autoclaving (121 °C for 20 min)	↑IVPD 7.2–11%	[26]
Irradiation	Irradiated at 0.5 and 1 kGy Irradiated at 0.5 and 1 KGy followed by cooking	↑IVPD 10–20% ↓IVPD 10–16%	[63]

**Table 5 molecules-28-05431-t005:** Thermal technologies used to reduce and eliminate antinutrients in faba bean seeds. Symbol ↑ donate improvement and ↓ indicate reduction.

Method	Conditions	Toxin Results	Reference
Cooking	Dehulling followed by soaking (1:10 *w*/*v*); cooking at 100 °C for 30 min	↓Phytic acid reduced by 13% ↓Trypsin inhibitors reduced by 15% ↓Tannins reduced from 63%	[26]
	Soaked for 4 h, followed by cooking at 95 °C for 1 h	↓Hemagglutinin activity reduced by 98% ↓Phytic acid reduced by 19% ↓Oxalate content reduced by 31%	[24]
Extrusion	Feeder at 383 and 385 g/min with moisture content at constant 25%. Outlet temperatures were 152 and 156 °C	↓Trypsin inhibitors reduced by 99% ↓Chymotrypsin inhibitor was reduced by 53% ↓a-Amylase inhibitor reduced by 100% reduction.	[107]
	Extrusion at 30–120 °C Cooking (soaked for 16 h before cooking (25–35 min)) Baked (193 °C for 35 min)	↓Saponins reduced by 13% ↓Phytic acid reduced by 50%	[34]
Autoclaving	Pre-treatment: dehulling and soaking followed by autoclaving at 1.5 × 10^6^ Pa (121 °C) for 20 min	↓Phytic acid reduced by 23–39 ↓Trypsin inhibitors reduced by 22–50% ↓Tannins reduced by 65%	[26]
Microwave cooking	Pre-treatment: soaking and dehulling followed by microwave cooking for 6 min	↓Trypsin inhibitors reduced by 8–52% ↓Tannins reduced by 0.5–58%	[26]
Heat treatment	Protein concentrate (95 °C for 15 min)	↓Relative trypsin activity reduced by 98%	[125]
	Boiling seeds for 0.33 h at 121 °C Roasting seeds for 0.17 h at 120 °C	↓Vicine reduced by 30% and convicine by 61% ↓Vicine reduced by 12% and convicine by 40%	[131]
Irradiation	Treatment: 0.5 and 0.1 KGy at a dose rate of 3.2 kGy/h at 25 °C, followed by cooking	↓Tannins reduced by 53% Phytic acid increased from 7.2 to 7.47 at 0.5 kGy but reduced at 6.5 mg/g at 1.0 kGy	[97]
Soaking–cooking	Soaked for 8 h; cooked in boiling water	Phytic acid (mg/100 g) decreased from 183.65 to 153.44 Tannin content (mg/100 g) reduced from 1120 to 50	[132]
Soaking	Solid: liquid (1:5 *w*/*w*) Time: 4 h at room temperature	↓Hemagglutinin activity reduced by 0.5–5% ↓Phytic acid reduced by 1–3% ↓Oxalate reduced by 17.40–37%	[23]
	1% acid (flow 0.5 mL/min) whole beans; 72 h at 50 °C	↓Vicine and convicine reduced by 100%	[108]
Autoclaving	Autoclaving after soaking (48 h) for 0.5 h at 121 °C	↓Vicine and convicine reduced by 50%	[108]

**Table 6 molecules-28-05431-t006:** Influence of various technologies on faba bean seeds’ pharmacological activity. Symbol ↑ donate improvement and ↓ indicate reduction.

Technology	Processing Conditions	Compound Tested	Major Findings	Unit	Reference
Untreated	Raw faba bean	Total phenols	0.45	mg/g	[136]
	Immature seed	Total phenols	817–1337.82	mg/GAE/Kg	[73]
	Seeds	Total phenols Total flavonoids Total carotenoids Antioxidants DPPH ABTS FRAP	2.5 2 2.1 74.71 2.35 70.16	mg/g μg/mL µg/mL µg Fe (III)/g	[148]
	Young leaves	Total phenols Total flavonoids DPPH	53.59–55.03 62.41–64.93 73.23–74.17	mg/GAE/g d.w mg/CE/g d.w mg VCE/g d.w	[149]
	Old leaves	Total phenols Total flavonoids DPPH	42.68–44.3 43.46–43.90 53.83–55.27		
Soaked–cooked	Whole seeds (soaked for 4 h, followed by cooking for 1 h)	Total polyphenols	−54% reduction		[63]
	Split beans		−47% reduction		
Enzymatic treatment	Alcalase-treated protein concentrate for 8 min	Antioxidants FRAP ORAC	FRAP increased by 6–36% ORAC increased by 2%		[146]
	Protein hydrolysates with different enzymes	Antioxidant ABTS DPPH	ABTS increased by 14,800% DPPH increased by 460%		[147]
Solvent extraction	Acetone extract	Total phenols Flavonoids	33.87 27.93		[150]
Germination	Time: 96 h; followed by digestion till the end of the small intestine	Antioxidants ORAC ABTS	ORAC increased by 112% ABTS increased by 95%		[27]
Bioprocess	Pasta replaced with 35% faba bean flour	Total phenols Total flavonoids Antioxidant activity	Phenol content increased by 190% Flavonoids increased by 74% Antioxidant content increased by 18%		[151]
Dry heating	100 °C for 15–60 min	TPC TFC Antioxidant activity	↓TPC after 15 min ↓TFC reduced ↓Antioxidant activity reduced		[149]
Steaming	Time: 15–60 min	TPC TFC Antioxidant activity	↓TPC after 15 min ↓TFC ↓Antioxidant activity reduced		

## Data Availability

The data generated during the current study are available from the corresponding author upon reasonable request.

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
