# Peer review of "Faba Bean Processing: Thermal and Non-Thermal Processing on Chemical, Antinutritional Factors, and Pharmacological Properties"

_molecules, 2023, doi:10.3390/molecules28145431_

Round 1
Reviewer 1 Report
This review manuscript was presented well. However, many major/minor errors need to be solved critically for consideration of the next stage.
In title, each word's first character should be capitalised per the journal guidelines.
The mechanism is poorly addressed for Thermal and non-thermal processing on a specific property. The authors need to highlight some figures. Also, discuss its chemistry of point of view.
Line 19, remove the comma.
Line 59: “The presence of”
Give the global research trends related to this study.
Line 105: “In Europe” instead of n Europe
Carbohydrates should be 2.3 and minerals 2.4
Line 203: The sentence is not correct.
Reframe the sentence in line 229
Line 235-241: Biological value has been explained in one line but then how PER is found has been written. It would be better if you could explain about PER in 1 line and also please check the last line.
Where is figure 2? It seems missing. Check it.
Table style is not ok as per journal guidelines. Double check it.
IVPD – Full form
In vitro- Italics
Where is Table 2. Revise carefully.
In Table 3: Everywhere it has been written respectively in results. It seems you have copied from some papers and pasted it. Respectively with respect to what. Change the whole table and write the results in a meaningful way.
Table 4: reduction
Line 665: verities: varieties
Provide Reference for Fig 3 if you directly copied from previous publication sources. Fig. 3. Resolution needs to be increased.
In tables 3. conditions information are too much write it preciously for readers compliance.
Two table are written as “Table 4”. One is non thermal and other thermal. Please change that.
Table 4 does “Toxin reduction” section doesn’t seem to be presented in a scientific way. Try to minimise the repeated words and better not to give so much data
Table 4: Are the technologies you have mention non thermal? Extraction, Soaking, Extraction, Fermentation?
In table of thermal techniques, you have written “High Pressure Processing” Is it a thermal technique?
Table 6: It is about various technologies but it is again giving the readers the same technology which have already been mentioned in the previous tables like: Germination, Soaking, Fermentation.
Make a future perspective in a different section. Preferable after conclusion.
Delete the very old and unnecessary references. Max. ref. should be withing 5 years.
All the 3 tables need lot of corrections to make the paper in an acceptable form.
Write the conclusion in a short and concise manner in a single paragraph.
Check the English writing thoroughly throughout the manuscript
Extensive editing of English language required
Author Response
REVIEWER 1 COMMENTS AND AUTHORS RESPONSE
This review manuscript was presented well. However, many major/minor errors need to be solved critically for consideration of the next stage.
Thank you very much for the kindly reviewing the paper and the corrections proposed. Corrections have been made to the best of knowledge.
In title, each word's first character should be capitalised per the journal guidelines.
Corrections have been made.
The mechanism is poorly addressed for Thermal and non-thermal processing on a specific property. The authors need to highlight some figures. Also, discuss its chemistry of point of view.
Corrections have been. Selected figures have been discussed in specific sections and highlighted. Line 632-640; 619-624; 663-667 etc. all other additional explanation has been highlighted with blue.
Line 19, remove the comma.
Correction done.
Line 59: “The presence of”
Corrections done.
Give the global research trends related to this study.
A graphical representation has been shown to summarize the global research trend over the years in Fig 2.
Line 105: “In Europe” instead of n Europe
Corrections have been made.
Carbohydrates should be 2.3 and minerals 2.4.
Corrections done.
Line 203: The sentence is not correct.
Sentence has been corrected.
Reframe the sentence in line 229.
Sentence has been reframed.
Line 235-241: Biological value has been explained in one line but then how PER is found has been written. It would be better if you could explain about PER in 1 line and also, please check the last line.
Line 290-291. PER has been explained briefly
Where is figure 2? It seems missing. Check it.
Figure has been added and checked throughout.
Table style is not ok as per journal guidelines. Double check it.
Table style has been formatted and redone.
IVPD – Full form
Corrected
In vitro- Italics
Where is Table 2. Revise carefully.
Table 2 has been included. Thanks for correction.
In Table 3: Everywhere it has been written respectively in results. It seems you have copied from some papers and pasted it. Respectively with respect to what. Change the whole table and write the results in a meaningful way.
Table 3 has been revised accordingly.
Table 4: reduction
Corrections have been made.
Line 665: verities: varieties
Corrections made.
Provide Reference for Fig 3 if you directly copied from previous publication sources. Fig. 3. Resolution needs to be increased.
Fig 3 has been updated and modified with reference.
In tables 3. conditions information is too much write it preciously for readers compliance.
Table 3 has been modified accordingly.
Two table are written as “Table 4”. One is non thermal and other thermal. Please change that.
Changes have been made.
Table 4 does “Toxin reduction” section doesn’t seem to be presented in a scientific way. Try to minimise the repeated words and better not to give so much data.
Table has been thoroughly modified.
Table 4: Are the technologies you have mention non thermal? Extraction, Soaking, Extraction, Fermentation?
Corrections have been made to the best of our understanding.
In table of thermal techniques, you have written “High Pressure Processing” Is it a thermal technique?
Corrections have been made.
Table 6: It is about various technologies, but it is again giving the readers the same technology which have already been mentioned in the previous tables like: Germination, Soaking, Fermentation.
Table 6 has been modified accordingly. Although technology may be the same; effect on antinutrients and polyphenols was separated to give readers a better understanding. Otherwise, the data will be too much on a single table.
Make a future perspective in a different section. Preferable after conclusion.
This has been done and included.
Delete the very old and unnecessary references. Max. ref. should be withing 5 years.
All the 3 tables need lot of corrections to make the paper in an acceptable form.
All tables have been modified appropriately. And again thank you very for the valuable corrections.
Write the conclusion in a short and concise manner in a single paragraph.
Corrections have been done.
Check the English writing thoroughly throughout the manuscript
This have been checked and corrections made accordingly
Reviewer 2 Report
In this review, a large number of papers was collected from a wide range of fields of faba bean processing, and was introduced according to experimental purposes. So, this review would help people who want to know what kind of studies have been performed in the field of faba bean processing in the last few years. On the other hand, the references include many reviews, which could be one of the reason why this review looks superficial. Thus it looks like this manuscript is just a list of reviews and papers.
Author Response
REVIEWER 2 COMMENTS AND AUTHORS FEEDBACK
In this review, a large number of papers was collected from a wide range of fields of faba bean processing and was introduced according to experimental purposes. So, this review would help people who want to know what kind of studies have been performed in the field of faba bean processing in the last few years. On the other hand, the references include many reviews, which could be one of the reasons why this review looks superficial. Thus, it looks like this manuscript is just a list of reviews and papers.
Thank you for the comments. Only few review papers were highlighted to shade light on areas that have been done in other aspects related legumes.
Round 2
Reviewer 1 Report
The authors improved the manuscript. Take copy right permission for all the copied figures from the previous publication. Make sure about this issue.
Minor editing of the English language required